# Distributed learning effectiveness in medical image analysis when trained with limited datasets

**Raissa Souza**[1,2]                                  RAISSA.SOUZADEANDRAD@UCALGARY.CA

**Pauline Mouches**[1,2]                               PAULINE.MOUCHES@UCALGARY.CA

**Matthias Wilms**[1,2,3]                              MATTHIAS.WILMS@UCALGARY.CA

**Anup Tuladhar**[1,2]                                 ANUP.TULADHAR@UCALGARY.CA

**Sönke Langner**[4]                                   SOENKE.LANGNER@MED.UNI-ROSTOCK.DE

**Nils D. Forkert**[1,2,3,5]                           NILS.FORKERT@UCALGARY.CA

[1] *Department of Radiology, University of Calgary, Calgary, Canada*

[2] *Hotchkiss Brain Institute, University of Calgary, Calgary, Canada*

[3] *Alberta Children's Hospital Research Institute, University of Calgary, Calgary, Canada*

[4] *Institute for Diagnostic Radiology and Neuroradiology, Rostock University Medical Center, Germany*

[5] *Department of Clinical Neurosciences, Cumming School of Medicine, University of Calgary, Calgary, Canada*

## Abstract

Federated learning (FL) is a cutting-edge method for distributed learning used in many fields, including healthcare. However, medical centers need sufficient local data to train local models and participate in an FL network, which is often not feasible for rare and pediatric diseases or small hospitals with limited patient data. As a result, these centers cannot directly contribute to FL model development. To address this issue, this work explores the effectiveness of a different approach called the travelling model (TM). Specifically, this work evaluates the performances of FL and TM when only very small sample sizes are available at each center. Brain age prediction was used as an example case for comparison in this work. Our results indicate that the TM outperforms FL across all sample sizes tested, particularly when each center has only one sample.

**Keywords:** Federated Learning, Travelling Model, Distributed Learning.

## 1. Introduction

The main technique used today for implementing distributed machine learning is federated learning (FL) (McMahan et al., 2016), where multiple models are iteratively trained in parallel at each center and are periodically aggregated at a central server. However, FL's performance is known to depend on the size of the datasets available at each center and often performs poorly when the dataset size is very small. In that case, sequentially training a single model at medical centers (known as travelling model (TM) (Souza et al., 2022a)) may be more suitable, as the single model iteratively sees all datasets. This study aims to compare FL and TM approaches for cases where centers can only provide very small samples, such as rare diseases.

To test and systematically evaluate those approaches in a medical image analysis setup, we use brain age prediction based on magnetic resonance imaging (MRI) data as an example

case. Although such data is widely and freely available, we chose this application scenario because the dataset used was acquired in a controlled way, which allows us to focus only on the effect of sample size, eliminating biases during model evaluation. Although brain age prediction is not a raree disease, we believe that the findings from this study will be applicable to other problems with limited data.

## 2. Material and Methods

### 2.1. Dataset

The data used in this study consists of morphological brain features extracted using Fast-surfer (Henschel et al., 2020) from a subset of the Study of Health in Pomerania (SHIP) (Volzke et al., 2011) consisting of 2025 cross-sectional T1-weighted MRI brain scans of pre-dominantly healthy adults aged between 21 and 82 years (mean: $50 \pm 13$ years). Training sets were divided into subsets to represent independent centers. These subsets varied in the number of samples per center, ranging from large to very small, specifically 20, 10, 5, 2, and 1 were used. To prevent selection-biased conclusions from being drawn from a single data split, ten Monte-Carlo cross-validation iterations were performed.

### 2.2. Models and Evaluation

A multi-layer perceptron was used for brain age prediction based on the extracted FreeSurfer features, consisting of two hidden layers with 256 and 128 neurons using ReLU activations, respectively, and an output layer with one neuron using a linear activation.

To ensure a fair comparison, all models (central model, federated learning model, and travelling model) were initialized with the same random weights and trained using the Adam optimizer with an initial learning rate of 0.01 for a total of 200 epochs. After the first ten epochs, an exponential learning rate decay of -0.1 was applied for every subsequent epoch.

For the FL model, evaluations were conducted using 20, 40, and 200 rounds, with each local model trained for ten, five, and one epoch(s), respectively, totaling 200 epochs in all cases. The TM was evaluated with the same number of rounds and epochs per round, and centers were visited in the same random order for all cycles to avoid any bias from varying travel orders.

The mean absolute error (MAE) was used for quantitative evaluation, as is standard in brain age prediction literature (Nam et al., 2020). This was calculated by computing the absolute difference between the predicted brain age and the known chronological age, with lower MAE values indicating better prediction performance.

## 3. Results

The central learning model produced the best MAE of 5.99 years, which falls within the range of previously published models that used similar tabulated data (Nam et al., 2020).

Figure 1 shows the average MAE across ten Monte-Carlo cross-validation iterations for the 18 distributed learning scenarios analyzed. These scenarios varied in the number of samples per site (1, 2, 5, 10, and 20) and the number of training rounds (20, 40, and 200).

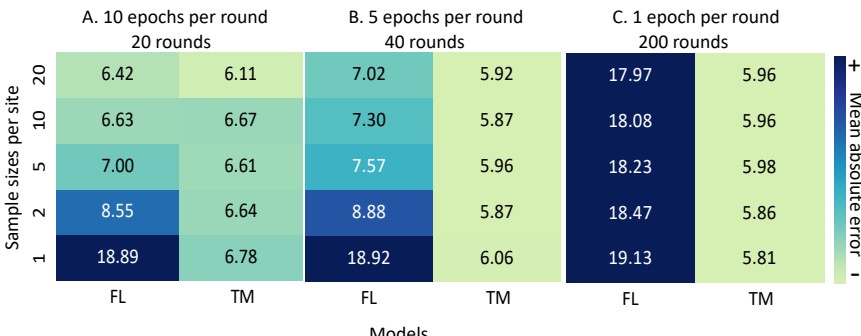

Figure 1: Federated learning model (FL) and travelling model (TM) performance across different sample sizes and rounds. To ensure consistency, the models were trained by seeing the entire dataset 200 times, with the number of epochs per site being controlled.

Figure 1 demonstrates that the FL model's performance declined as the sample size decreased, regardless of the number of rounds. For example, for both 20 and 40 rounds, the experiment with a single subject per center resulted in the worst performance (MAE of approximately 18.9 years). In contrast, the performance of the TM was relatively unaffected by small sample sizes (MAE of 6.78 years for 20 rounds and MAE of 6.06 years for 40 rounds with a single subject per center). The travelling model consistently produced results comparable to the central learning model (MAE of 5.99 years) throughout the experiments.

The results also revealed that both the distributed learning models' ability to learn meaningful relationships from the data was influenced by the number of epochs the model was trained per round. Figure 1A-C illustrates how the FL model's performance deteriorated as the number of epochs trained per round decreased. The model produced higher errors when trained for fewer epochs per round. In contrast, the TM's performance improved when the model was trained at each center for fewer epochs per round. Although the effect was less pronounced, it was statistically significant (two-tailed paired t-test $p < 0.009$).

## 4. Conclusion

This work evaluated the effectiveness of travelling models compared to federated learning models for medical imaging analysis with very small sample sizes. Our findings indicate that travelling models perform better than federated learning models, regardless of the sample size. Additionally, it was found that the travelling model can achieve similar results to central learning even with only one epoch of training at each site. As a result, travelling models may be a more suitable option for distributed learning with limited local datasets, creating new opportunities for applying machine learning models in rare diseases, pediatric research, and small hospitals contributing to distributed learning setups[1].

---

1. This short paper discusses the essentials of the work published in (Souza et al., 2022b)

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
