# OpenReview forum: "Distributed learning effectiveness in medical image analysis when trained with limited dataset"
_MIDL.io/2023/Short_Paper_Track — MIDL 2023 Short paper track Poster_

### Official Review · Reviewer_5xqt · 2023-04-21
**More convincing experimental setup needed**

**Rating:** 4
**Confidence:** 3

**Review:**

The paper proposes to iteratively train a model on data from different centers, as opposed to the federated learning scenario. The experiments are done with a brain age prediction task on data from a single center, the proposed model outperforms the federated learning scenario.

Pros:
* Combining data across centers is important
* Straightforward approach

Cons:
* The use case (task + single center data) feels too artificial, for brain age prediction there are many publicly available datasets that could be used, results on rare disease prediction (like the motivating example) would be more relevant
* What is the number of centers? How is the order of the centers decided?

---

### Official Review · Reviewer_xN7b · 2023-04-22
**An evaluation of federated learning vs travelling models for distributed training**

**Rating:** 7
**Confidence:** 4

**Review:**

The paper addresses empirically an important question for medical imaging, namely distributed learning with limited data size.

Pros:
- several hyperparameters explored and showing catastrophic effects of low sample size, respectively few local epochs
- Useful results for other studies

Cons:
- Open source code is not mentioned
- the intro conflates federated learning (a generic approach) with FedAvg (a specific algorithm). However other FL approaches, in particular cyclic learning (Sheller et al BrainLes 2018), may perform very differently. I understand this is a small paper, but the intro should at least acknowledge other algorithms.
- The implementation of the travelling model should be explained in slightly more detail - is it just retraining for each partition of the dataset, using weights trained on the previous partition as initialisation?